# Minimal transcriptional regulation of horizontally transferred photosynthesis genes in phototrophic bacterium *Gemmatimonas phototrophica*

Karel Kopejtka,[1] Jürgen Tomasch,[1] Sahana Shivaramu,[1] Mohit Kumar Saini,[1] David Kaftan,[1] Michal Koblížek[1]

**ABSTRACT** The first phototrophic member of the bacterial phylum *Gemmatimonadota*, *Gemmatimonas phototrophica* AP64[T], received all its photosynthesis genes via distant horizontal gene transfer from a purple bacterium. Here, we investigated how these acquired genes, which are tightly controlled by oxygen and light in the ancestor, are integrated into the regulatory system of its new host. *G. phototrophica* grew well under aerobic and semiaerobic conditions, with almost no difference in gene expression. Under aerobic conditions, the growth of *G. phototrophica* was optimal at 80 µmol photon $m^{-2}$ $s^{-1}$, while higher light intensities had an inhibitory effect. The transcriptome showed only a minimal response to the dark–light shift at optimal light intensity, while the exposure to a higher light intensity (200 µmol photon $m^{-2}$ $s^{-1}$) induced already stronger but still transient changes in gene expression. Interestingly, a singlet oxygen defense was not activated under any conditions tested. Our results indicate that *G. phototrophica* possesses neither the oxygen-dependent repression of photosynthesis genes known from purple bacteria nor the light-dependent repression described in aerobic anoxygenic phototrophs. Instead, *G. phototrophica* has evolved as a low-light species preferring reduced oxygen concentrations. Under these conditions, the bacterium can safely employ its photoheterotrophic metabolism without the need for complex regulatory mechanisms.

**IMPORTANCE** Horizontal gene transfer is one of the main mechanisms by which bacteria acquire new genes. However, it represents only the first step as the transferred genes have also to be functionally and regulatory integrated into the recipient's cellular machinery. *Gemmatimonas phototrophica*, a member of bacterial phylum Gemmatimonadota, acquired its photosynthesis genes via distant horizontal gene transfer from a purple bacterium. Thus, it represents a unique natural experiment, in which the entire package of photosynthesis genes was transplanted into a distant host. We show that *G. phototrophica* lacks the regulation of photosynthesis gene expressions in response to oxygen concentration and light intensity that are common in purple bacteria. This restricts its growth to low-light habitats with reduced oxygen. Understanding the regulation of horizontally transferred genes is important not only for microbial evolution but also for synthetic biology and the engineering of novel organisms, as these rely on the successful integration of foreign genes.

**KEYWORDS** Gemmatimonadota, anoxygenic photosynthesis, bacteriochlorophyll, horizontal gene transfer, transcriptomics

Address correspondence to Michal Koblížek, koblizek@alga.cz.

The authors declare no conflict of interest.

See the funding table on p. 12.

Life on planet Earth is sustained through the photosynthetic (PS) transformation of light into chemical energy (1, 2). The majority of primary biomass is produced by PS organisms (e.g., cyanobacteria, algae, and plants), which harvest light energy

using chlorophylls, fix inorganic carbon, and evolve oxygen. Alongside these oxygenic organisms, anoxygenic phototrophs harvest light using various forms of bacteriochlorophyll (BChl) and have been found so far in eight bacterial phyla: *Pseudomonadota* (*Proteobacteria*), *Bacteriodota* (*Chlorobiota*), *Bacillota*, *Chloroflexota*, *Gemmatimonadota*, *Acidobacteriota*, *Myxococcota*, and *Vulcanimicrobiota* (formerly known as *Candidatus* Eremiobacteriota or WPS-2).

An interesting group is represented by the phototrophic *Gemmatimonadota*. Its first cultured member *Gemmatimonas phototrophica* AP64[T] was isolated from a freshwater lake Tiān ér hú in the western Gobi Desert in Northern China (3). It contains unique PS complexes composed of the type-2 reaction center with two concentric rings of light-harvesting antennae (4, 5). *G. phototrophica* is a photoheterotrophic bacterium. It requires organic carbon, but light provides energy for its metabolism and stimulates its growth (6). Phylogenetic evidence suggests that *G. phototrophica* received the photosynthesis gene cluster (PGC) via horizontal gene transfer (HGT) from purple bacteria (3).

The early anoxygenic phototrophs evolved under anaerobic conditions during the Archean eon (7). Whereas the extant phototrophic *Bacillota* or *Bacteriodota* remained as strictly anaerobic species, other phototrophic groups evolved different strategies to cope with the modern oxic atmosphere. For example, purple bacteria (e.g., *Rhodobacter capsulatus* or *Cereibacter sphaeroides*) synthesize their PS apparatus and grow photoautotrophically only under anoxic conditions. At higher oxygen tension, they repress their pigment synthesis (8–10) and switch to heterotrophic metabolism (11). Another strategy was adopted by aerobic anoxygenic phototrophic (AAP) bacteria. These photoheterotrophs fully adapted to oxic conditions and harvest light energy and grow in the presence of oxygen. The synthesis of PS apparatus under illumination in the presence of oxygen, however, represents a risk of generation of reactive oxygen species (ROS) (12), potentially damaging proteins, lipids, or DNA (13, 14). Therefore, AAP species rapidly downregulate the expression of photosynthesis genes when illuminated (15–17), which consequently also stops BChl biosynthesis (18–20) and, hence, avoids ROS generation under aerobic conditions (21). Environmental studies confirmed this mechanism taking effect in the upper ocean or freshwater lake epilimnia, where AAP bacteria restricted their BChl *a* synthesis to the night period (19, 22). Thanks to this ability, AAP bacteria contribute by a large part (up to 20%–30%) to microbial communities in euphotic zones of freshwater and marine habitats, respectively (23–25).

Yet another situation was observed in *G. phototrophica*. It was shown that cells grown on agar produced BChl under semiaerobic conditions in both dark and light (6, 26). Recently, it was shown that when the original medium is supplemented with calcium, *G. phototrophica* can grow also in fully aerobic liquid cultures (27), which also maintains BChl synthesis.

We speculate that the unusual BChl synthesis regulatory pattern may be related to the fact that its photosynthesis gene were received via distant HGT. Thus, they may not be fully integrated into the host gene regulation machinery. A recent metagenomics study documented the presence of phototrophic *Gemmatimonadota* in many fresh and waste waters (28), which indicates that they live in environments with variable oxygen concentrations. Therefore, we decided to investigate how *G. phototrophica* changes its gene expression under different oxygen and light conditions. Using a series of cultivation growth experiments combined with transcriptome analyses, the presented study reveals how the bacterium regulates PGC expression under different light conditions or oxygen tension.

## RESULTS

### Heterotrophic growth and BChl *a* synthesis under aerobic and semiaerobic conditions

Purple bacteria downregulate their photosynthesis gene expression in the presence of oxygen. An important component of this regulation is PpsR aerobic repressor, which halts photosynthesis gene transcription (29–31). As *G. phototrophica* contains a gene

coding for this regulator in its PGC and it is a semiaerobic species, we first investigated how oxygen concentration influences the expression of its PS apparatus. Cells grown in agar manifested the most vigorous growth in a zone ca. 0.5 cm under the agar surface, clearly showing their preference for lower oxygen tension (Fig. 1A, left inset). Then, *G. phototrophica* was cultured in a liquid medium in the dark under semiaerobic (10% $O_2$) or aerobic (21% $O_2$) conditions, and its growth and pigmentation were monitored. The semiaerobic cultures reached on average 20% higher density compared with aerobic cultures between days 3 and 9 (Fig. 1A). However, the pigmentation assayed as BChl *a* per protein ratio (w:w) was the same for both treatments (Fig. 1A, right inset), which signalizes the absence of aerobic repression in *G. phototrophica*.

## Transcriptome response following the dark-to-light switch under semiaerobic and aerobic conditions

As there was no apparent impact of the oxygen concentration on cellular pigmentation in cultures grown in the dark, we assayed whether *G. phototrophica* changes its photosynthesis genes under different oxygen concentrations. Cultures grown in the dark were split into two sets of subcultures kept under semiaerobic or aerobic conditions. Genes in the PGC showed a similar high expression (above the median for all genes) under both oxygen regimes (Fig. 1B). Further, one set of bottles was exposed to light intensity of 100 µmol photon m$^{-2}$ s$^{-1}$ while the control set was kept in the dark. Samples were taken before and 2 and 8 hours after the shift to light. We considered only genes with a fold change (FC) >2 and *P*-value <0.05 to be differentially expressed. To identify the influence of light, we first compared the dark–light transitions for both oxygen concentrations separately (Fig. S1; Table S1). The response to illumination was similarly weak with only 35 and 41 differentially expressed genes (DEGs) under semiaerobic and aerobic conditions, respectively. Only the *puf* genes showed temporary weak repression upon light under both conditions alike. The same was seen for the *hemEFG* genes, coding for the biosynthesis of the BChl-precursor protoporphyrin IX. The only notable difference between both time series was the 12-fold downregulation of ferrochelatase *hemH* after 8 hours in light only at 10% $O_2$. This enzyme represents the branching point into heme biosynthesis.

To identify the influence of oxygen, we compared the gene expressions between cultures grown under semiaerobic and aerobic conditions according to the sampling points in the dark and light. We obtained 41 DEGs that were grouped into four clusters (Fig. 1C; Table S2). Cluster 1 was characterized by a consistently high expression clearly triggered by the lowered oxygen tension. It contained five genes coding for the cbb3 type cytochrome c oxidase with high affinity to $O_2$, the oxygen-independent coproporphyrinogen III oxidase (*hemN*), universal stress protein, and a nitronate monooxygenase. Cluster 2 contained genes with a weaker upregulation under semiaerobic conditions and different dynamics in the course of the transition from dark to light. Cluster 3 comprised already mentioned ferrochelatase (*hemH*) and a neighboring gene, which were both reduced under semiaerobic conditions in the dark. Finally, cluster 4 consisted of genes weakly downregulated under semiaerobic conditions. Of note were the high number of genes involved in transport as well as regulators and sensors. In accordance with the previous comparison (light vs dark), photosynthesis genes were not among DEGs in response to $O_2$ concentration.

## Influence of light intensity and illumination regime on aerobic growth, BChl *a* synthesis, and photosynthesis

The second factor, which influences photosynthesis gene expression in almost all PS organisms, is light intensity. Therefore, we decided to first analyze the effect of light intensity on growth, BChl *a* synthesis, and PS activity. The aerobic cultures were grown at 0, 30, 80, 200, and 500 µmol photon m$^{-2}$ s$^{-1}$ delivered either continuously or using a 12-hour light–12-hour dark regime. During 10 days of cultivation, compared to controls grown in the dark, light intensities of 30 and 80 µmol photon m$^{-2}$ s$^{-1}$ enhanced the

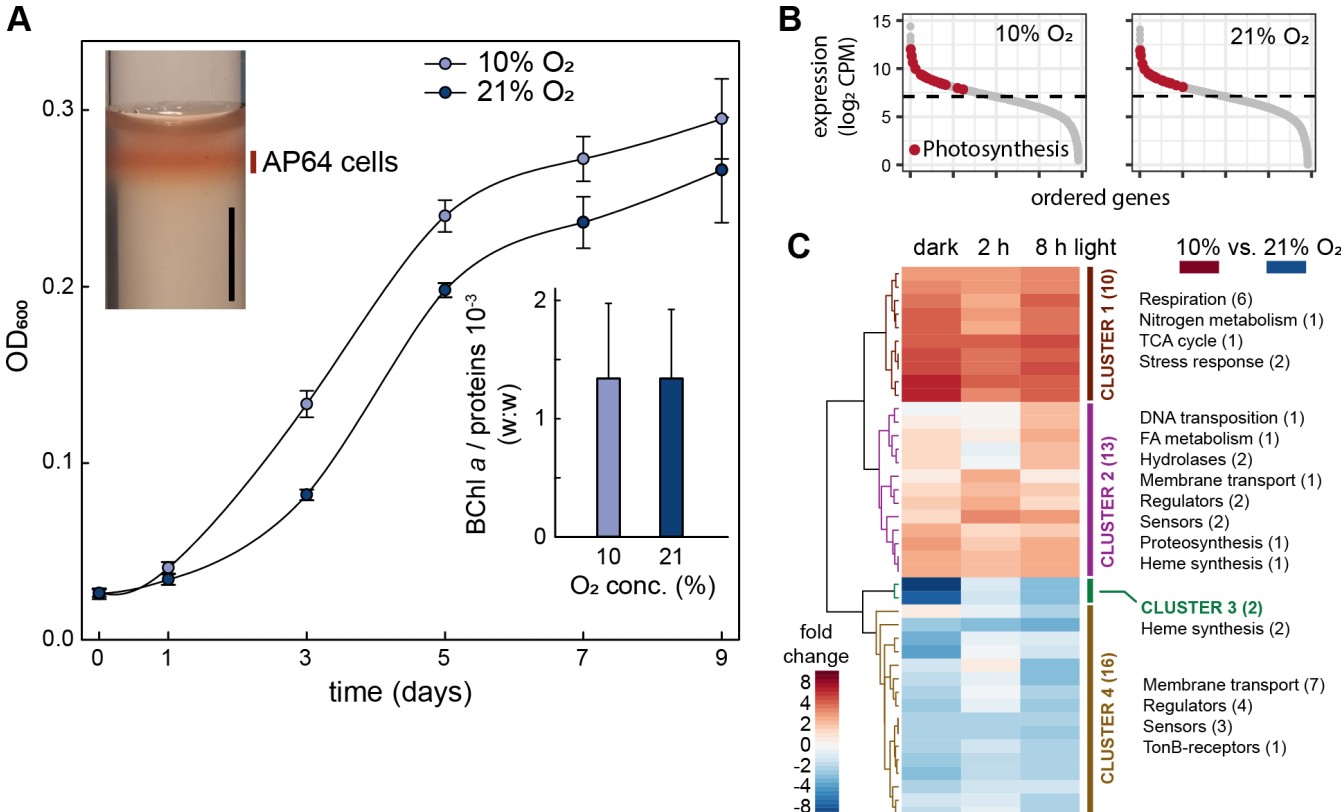

**FIG 1** Cultivation under different oxygen concentrations. (A) Heterotrophic growth under semiaerobic (10% O₂) and aerobic (21% O₂) conditions. The mean values and standard deviations from three parallel biological replicates for each treatment (defined in the graphical legend) are shown. (Left inset) Cells grown inside the agar clearly prefer a semiaerobic environment. The black vertical scale bar represents 1 cm. (Right inset) BChl *a* per protein ratio. For each oxygen concentration, the mean value calculated as an average of days 3, 5, 7, and 9 is shown. (B) Expression of photosynthesis genes in the dark (red) under 10% and 21% oxygen, compared to the whole transcriptome (gray). The dashed line shows median expression. (C) Transcriptional response to the dark-to-light shift under semiaerobic and aerobic conditions. Heatmap visualizations of the difference in fold changes (FCs) of differentially expressed genes (DEGs) between cells grown under semiaerobic and aerobic conditions. Dark-adapted cells were illuminated with 100 µmol photon m⁻² s⁻¹ of light, and samples were taken at 2 and 8 hours after the shift to light. For normalization, samples kept in the dark were used. DEGs are clustered into four main groups (Clusters 1–4) marked by vertical bars. Groups of genes with common functions are shown. Numbers in brackets show the number of genes represented in each gene cluster/group. Hypothetical proteins and proteins with domains of unknown function are not shown. Cut-off values used for the analysis: *P*-value =0.05; FC > 2. CPM: counts per million reads; FA, fatty acid; OD, optical density.

growth of *G. phototrophica* (Fig. 2). This positive effect was more pronounced under continuous illumination (Fig. 2, lower panel). However, at the highest light intensity of 500 µmol photon m⁻² s⁻¹, the growth was significantly hindered or completely arrested under both illumination regimes. To monitor BChl *a* biosynthesis, time series of samples from days 5, 7, 9, and 10 were taken and analyzed, and an average BChl *a* per protein (w:w) ratio for each light intensity and both illumination regimes was calculated (Fig. 2, insets). Under both illumination regimes, the highest average BChl *a* per protein (w:w) ratio was documented in cultures grown at a light intensity of 30 µmol photon m⁻² s⁻¹. At higher light intensities, BChl *a* production was reduced. Cultures grown at the highest light intensity (500 µmol photon m⁻² s⁻¹) under a 12-hour dark–12-hour light regime had approx. 3.7 times lower pigmentation when compared to cultures grown in the dark (7.70 ± 2.71 × 10⁻⁴ vs 2.85 ± 0.35 × 10⁻³ BChl *a* per protein, w:w, respectively) (Fig. 2, upper panel, inset). Due to the detrimental effect of high light, we were not able to detect any BChl *a* in cultures incubated at the highest light intensity under continuous illumination (Fig. 2, lower panel, inset).

Cultures grown at continuous light adapted their PS apparatus in response to light intensity (Fig. S2). The yield of primary photochemical reactions estimated from BChl

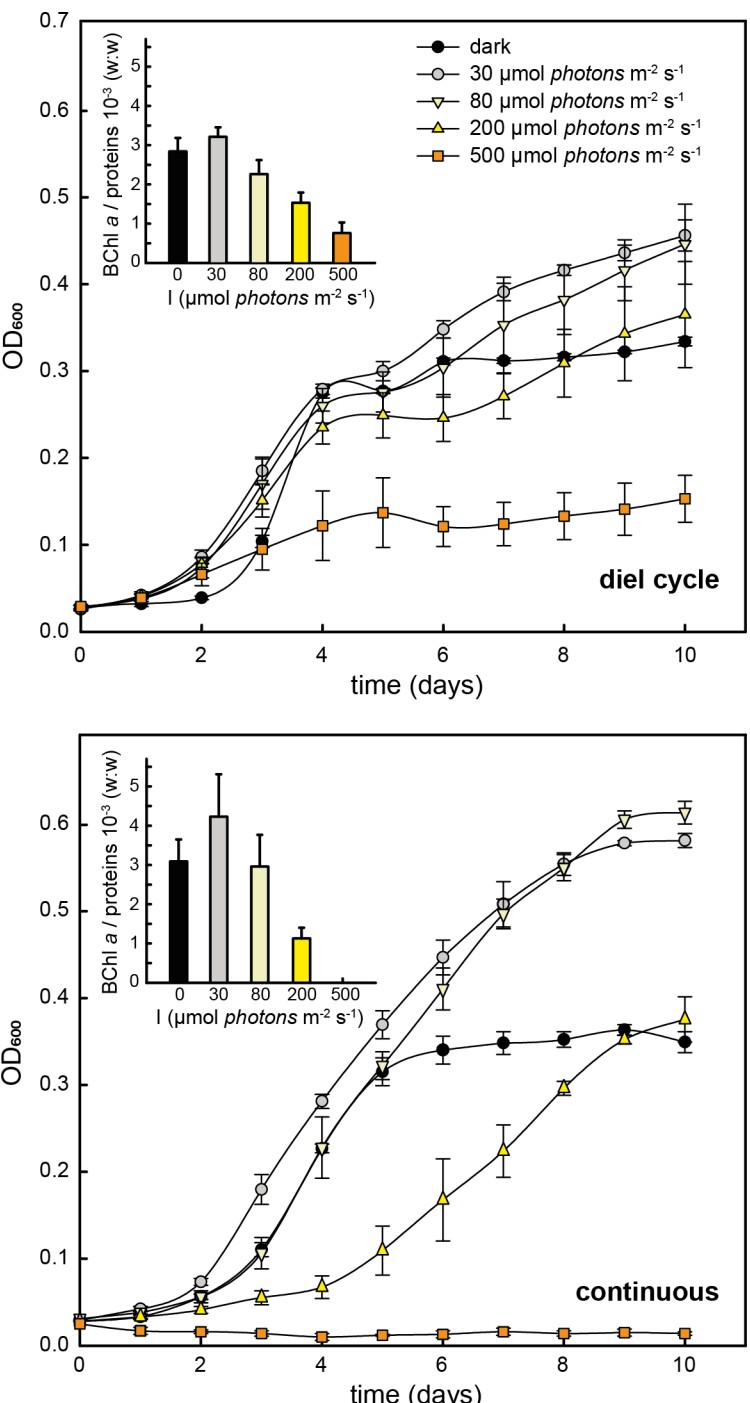

**FIG 2** Cultivation under different light intensities and illumination regimes. Cells were grown photoheterotrophically under aerobic (21% $O_2$) conditions. Upper panel: 12-hour dark–12-hour light regime. Lower panel: continuous illumination. The mean values and standard deviations from three parallel biological replicates for different light intensities (defined in the graphical legend) are shown. (Insets) BChl *a* per protein ratio. For each light intensity (I), the mean value calculated as an average of days 5, 7, 9, and 10 is shown.

fluorescence parameter $F_V/F_M$ was only slightly although significantly higher (Student's *t*-test, *P*-value =0.002) in cells grown in the light irrespective of its intensity $F_V/F_M = 0.68 \pm 0.01$, in comparison with the dark-adapted cells $F_V/F_M = 0.65 \pm 0.02$. The PS electron transfer approximated by the reopening rate of the reaction center with two rings of

light-harvesting antennae, $k_{re-open}$, responded non-linearly to the increasing light intensity in a similar way to that reported in *Dinoroseobacter shibae* (32). The functional antenna cross-section of the reaction center with two rings of light-harvesting antennae, $\sigma_{RC-dLH}$, decreased from 79 Å in dark-adapted cells to 66 Å in the cells grown at 500 µmol photon $m^{-2}$ $s^{-1}$. No connectivity of the reaction center with two rings of light-harvesting antennae, $J_{con}$, was observed.

## Transcriptome response following illumination with different light intensities under aerobic conditions

As there was only minimal effect at low light intensity, we decided to investigate the gene expression also at a higher light intensity of 200 µmol photon $m^{-2}$ $s^{-1}$. This intensity was chosen based on previous experiments (see Fig. 2) as suboptimal, but not strongly damaging. The cultures were grown aerobically in the dark until the mid-exponential phase and then illuminated for 8 hours with light intensities of 100 or 200 µmol photon $m^{-2}$ $s^{-1}$. Thus, cultures were compared under conditions with and without a reduction in pigment content and reduced growth, respectively. For control of the growth effect on gene expression in light-exposed samples, we used control samples kept in the dark. For the whole data set, we obtained 128 DEGs that were grouped into four main clusters with several subclusters (Fig. 3A; Table S3).

Under low light, the vast majority of the selected genes were not differentially expressed. The few exceptions were almost exclusively downregulated. DEGs in cluster 1 strongly decreased in expression under prolonged illumination. The strongest repression of approx. 10-fold was detected for a *YceI* family protein and a 2-oxoacid:acceptor oxidoreductase family protein. *YceI* family proteins are often associated with stress response, DNA repair, and environmental adaptation. The 2-oxoacid:acceptor oxidoreductase family is a group of enzymes that catalyze the transfer of electrons from 2-oxoacids (alpha-keto acids) to acceptor molecules. Subcluster 2a contained some genes of the PGC, *hemF*, and *hemG* that showed transient repression and de-repression during the first 4 hours in the light. Cluster 2b showed a transient downregulation only 4 hours after illumination. Only few solitary DEGs were more than fourfold upregulated under low light intensity, including those coding for a carboxypeptidase-like regulatory domain-containing protein, 4 hours, a phosphate ABC transporter permease subunit (*pstC*), and a PEP-CTERM sorting domain-containing protein 8 hours after the shift to light. Carboxypeptidases are enzymes that cleave amino acids from the C-terminus of proteins or peptides. Phosphate ABC transporters facilitate the uptake of phosphate from the external environment into the cell. Proteins containing the PEP-CTERM domain are often involved in biofilm formation in environmental bacteria associated with aquatic sediments and soils (33).

Higher light intensity had a stronger effect also on gene expression (Fig. 3A). Only genes in cluster 1 and subcluster 2b were very weakly regulated. Two hours after the shift to light, more than half (55%) of DEGs, in subclusters 2a and 2c, were transiently repressed with a gradual de-repression during the following 6 hours. Among the most downregulated genes was a gene coding for an enzyme (*bshB1*; EC: 3.5.1.-) crucial for bacillithiol biosynthesis, a cytochrome *c* oxidase, a 2-oxoacid:acceptor oxidoreductase (EC: 1.2.7.3, 1.2.7.11), and genes for beta and delta subunits of a formate dehydrogenase (EC: 1.17.1.9). In bacterial cells, bacillithiol helps to maintain redox homeostasis (34). Formate dehydrogenases are enzymes that catalyze the oxidation of formate to $CO_2$ while providing electrons for the respiratory chain. Most prominent among these transiently downregulated genes was the PGC (Fig. 3B). The strongest repression showed the operon coding for structural proteins. Similar, although weaker, repression patterns showed four operons with genes for BChl *a* synthesis. Other moderately downregulated photosynthesis genes were *acsF* and *lhaA*. The genes coding for the regulators *ppsR* and *ppaA* did not change in expression. The *hemEFG* operon, located next to the PGC, was also strongly downregulated.

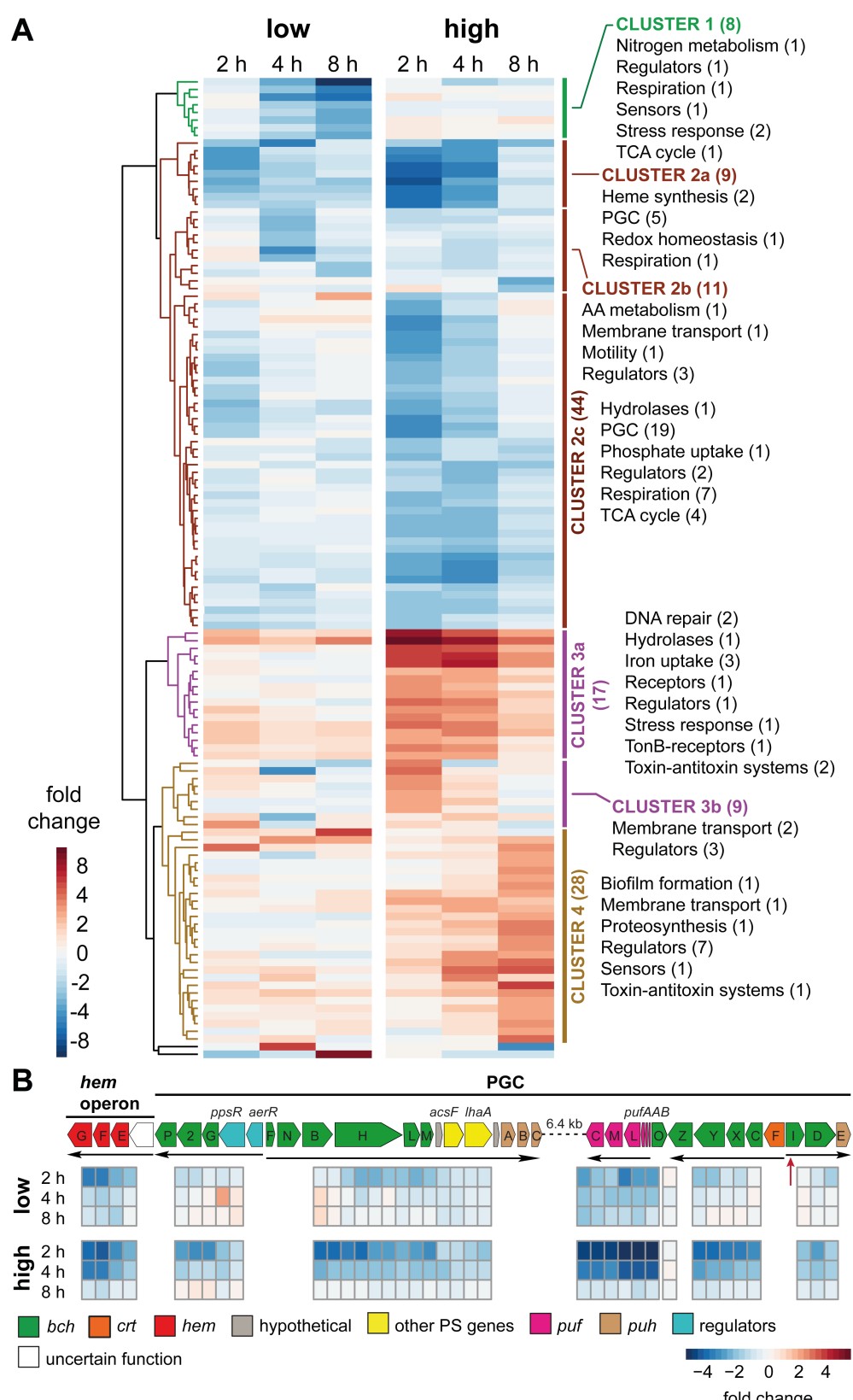

**FIG 3** Transcriptional response to the dark-to-light shift under aerobic conditions. (A) Heatmap visualizations of the difference in FCs of DEGs between dark-adapted cells illuminated with low (100 µmol photon m$^{-2}$ s$^{-1}$) or high (200 µmol photon m$^{-2}$ s$^{-1}$) light intensity. Samples were taken at 2, 4, and 8 hours after the shift to light. For normalization, samples kept in the dark (Continued on next page)

**FIG 3** (Continued)

were used. DEGs are clustered into four main clusters (Clusters 1–4) marked by vertical bars. Groups of genes with common functions are shown. Numbers in brackets show the number of genes represented in each gene cluster/group. Hypothetical proteins and proteins with domains of unknown function are not shown. Cut-off values used for the analysis: *P*-value =0.05; FC > 2. (B) Expression dynamics of PGC and the adjacent *hem* operon. Genes are colored based on their function. *bch* (green), bacteriochlorophyll biosynthesis genes; *crt* (orange), carotenoid biosynthesis genes; *hem* (red), heme biosynthesis genes; gray, genes coding for hypothetical proteins; *puf* (pink), genes encoding reaction center proteins; *puh* (brown), genes encoding reaction center assembly proteins; blue, regulatory genes; yellow, other PS genes; white, genes with uncertain function. Arrows mark involved operons. The putative binding site for PpsR is marked in red. No cut-off values were used for the analysis.

DEGs in clusters 3 and 4 increased in expression, but with different temporal dynamics (Fig. 3A): subcluster 3a was upregulated during the first 4 hours of illumination. It contained genes coding for the DNA photolyase, stress response, a TonB-dependent iron uptake system, and an alpha/beta fold hydrolase with a response regulator in one operon. The latter two genes showed the highest induction, up to 8.5-fold. Alpha/beta fold hydrolases are a broad class of enzymes associated with diverse physiological roles, including detoxification processes, maintenance of homeostasis, and signal transduction, but their role under these conditions is unclear. Subcluster 3b contained DEGs activated only transiently 2 hours after the shift to light. Genes in cluster 4 showed a gradual up to fivefold activation during the progression from dark to 8 hours of illumination. Approximately 25% of DEGs in this cluster were annotated as transcription factors or regulators.

## DISCUSSION

HGT represents one of the main mechanisms by which bacteria acquire new genes. However, it represents only the first step as the permanent integration of the newly acquired genes into the host cellular machinery requires functional and regulatory adaptation (35). The PGC synteny as well as phylogenetic evidence suggests that phototrophic *Gemmatimonadota* acquired their PGC horizontally from a purple bacterial ancestor (3, 36).

Phototrophic organisms have evolved various adaptation mechanisms that maintain the balance between light energy input and metabolic needs on the output. Anoxygenic phototrophs usually control their pigmentation—they produce more pigments under lower light and reduce pigmentation under higher light conditions. In anoxygenic phototrophs, this step is regulated already on the transcription level. Purple non-sulfur bacteria downregulate the expression of their photosynthesis genes in the presence of oxygen (37). In contrast, *G. phototrophica* grew and produced pigments under semiaerobic as well as aerobic conditions (Fig. 1A). Also, its PGC expression was not affected despite the presence of *ppsR* and *ppaA* genes (Fig. 1B). The only found change was the activation of the cbb$_3$ cytochrome *c* oxidase with high oxygen affinity (37, 38) and the downregulation of ferrochelatase under semiaerobic conditions. Ferrochelatase represents a key branching point between heme and BChl biosynthesis, and it was proposed to balance the flow of intermediates between these important pathways in *Cereibacter sphaeroides* (39).

In the presence of oxygen, most AAP bacteria perform pigment biosynthesis only in the dark as their PGC is repressed by light (15, 16, 18, 40). In contrast, *G. phototrophica* remains fully pigmented even when grown under continuous low light on agar plates (6) or in liquid cultures (Fig. 2, lower panel, inset). In our experiments, we observed transcriptional downregulation of the PGC at higher light intensities, but it was only transient and recovered after 8 hours (Fig. 3B). Thus, the permanent repression mechanism as previously characterized for proteobacterial AAP bacteria (15, 16, 40, 41) was not observed. The ability to synthesize BChl under aerobic conditions and light resembles the recently described phototrophic bacterium *Sediminicoccus rosea* isolated from Iceland (42). A comparison of the expression of photosynthetic gene *pufM* and regulator *ppsR* revealed distinct patterns for different AAPs (Fig. 4). In marine *D.*

*shibae* and freshwater *Sphingomonas alpina*, *pufM* is strongly repressed while *ppsR* is constitutively expressed. In *D. shibae,* illumination induces blocking by the transcriptional repressor through the release of PpsR-bound proteins LdaP and PpaA, resulting in the downregulation of its target genes (43). In both *G. phototrophica* and *S. rosea,* the PGC genes are transiently repressed, but to a dramatically different extent (2-fold vs 32-fold). In *S. rosea,* the expression of *ppsR* (and antirepressor gene *ppaA*) followed the same pattern and intensity as genes supposedly regulated by the PpsR-PpaA system (42). An auto-regulatory reduction of the repressor concentration could explain the transient repression of the targets. On the contrary, in *G. phototrophica,* the expression of *ppsR* did not change in response to light, suggesting a different mechanism controlling the transient repression of the PGC. Moreover, we identified only one binding site for PpsR inside the PGC, at the beginning of the *bchID-puhE* operon (Fig. 3B).

Besides the regulation of the PS apparatus, other light adaptation mechanisms exist. One of the most important is protection against ROS to prevent oxidative damage (12–14). In *G. phototrophica,* the ROS defense includes catalase, glutathione and glutathione peroxidase, superoxide dismutase [Mn], and thioredoxin and thioredoxin reductase. Similar to *S. rosea* (42), the genes coding for these enzymes were constitutively transcribed without significant changes under any conditions tested. However, a certain stress response is observed after the transition to high light, namely, fast activation of two photolyases repairing damaged DNA (Fig. 5). PS organisms also need to regulate the cellular metabolism that uses the harvested light energy to prevent any metabolic imbalance. Here, we observed that *G. phototrophica* downregulates the expression of its respiratory genes and genes of the TCA cycle (Fig. 5). On the other hand, proteosynthesis and proteolysis genes were gradually upregulated (Fig. 5), which may result in increased protein turnover.

*G. phototrophica* acquired the PS apparatus via HGT from a purple bacterial ancestor (3). Later, it evolved a novel type of PS complex with a very large light-harvesting antenna (5), which allows an effective harvesting of energy under low light conditions. Here, we present evidence that the genetic control of the PGC was not transferred from the original host, and *G. phototrophica* has not evolved any specific regulation of its PS apparatus on its own. All these characteristics constrain its growth to low light as higher light intensities may cause cellular damage (see also Fig. 2). Thus, the physiology of *G. phototrophica* seems to lie between microaerophilic purple non-sulfur species and AAP bacteria, as discussed by Zeng et al. (36). This suggests that it may thrive in environments characterized by reduced oxygen and lower light. One such niche would be the hypolimnion in stratified freshwater lakes. Indeed, the common presence of photoheterotrophic *Gemmatimonadota* was found in metagenomes collected in the hypolimnia of five freshwater lakes in central Europe. Based on 16S rRNA gene frequency, this clade represented 0.1%–1% of all bacteria (44). Photoheterotrophic *Gemmatimonadota* may be also present in upper illuminated sediment layers as documented before in Lake Taihu, China (45).

## MATERIALS AND METHODS

### Cultivation and sampling

*G. phototrophica* AP64[T] (=DSM 29774) was grown in an optimized liquid medium described earlier (27) at 25°C. The medium contained ($L^{-1}$) 0.5 g yeast extract, 0.5 g peptone, 0.3 g pyruvate, 0.5 g glucose, 0.5 g soluble starch, 0.3 g $K_2HPO_4$, 20 mg $CaCl_2$, 1 mL of modified SL8 trace metal solution ($mL^{-1}$: 190 µg $CoCl_2 \cdot 6H_2O$, 5.2 mg $Na_2$-EDTA, 24 µg $NiCl_2 \cdot 6H_2O$, 17 µg $CuCl_2 \cdot 2H_2O$, 70 µg $ZnCl_2$, 20.3 mg $MgCl_2$, and 62 µg $H_3BO_3$), and 1 mL of vitamin solution ($mL^{-1}$: 200 µg B1, 20 µg B3, 10 µg B7, and 10 µg B12); pH 7.3. Cultures for all experiments were grown in triplicates and monitored by turbidity measurements at 600 nm using the DEN-600 photometer (Biosan SIA, Latvia). At the beginning of each experiment, the inoculum (approx. $OD_{600} = 0.2$) was diluted in 100 mL of a fresh medium to approx. $OD_{600} = 0.03$. Aerobic (21% $O_2$) cultures were grown in

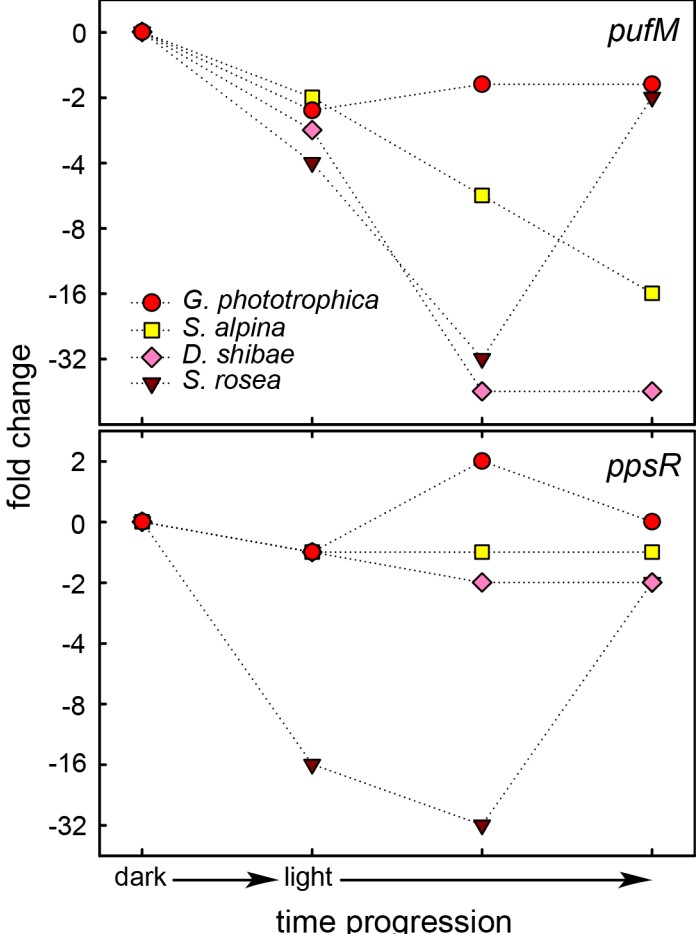

**FIG 4** Comparison of temporal expression patterns of *pufM* and *ppsR* genes in different AAP bacteria. Fold changes as recorded during the dark-to-light shift in *G. phototrophica* AP64 (this study), *Sediminicoccus rosea* KRV36 (42), *Sphingomonas glacialis* AAP5 (16), and *Dinoroseobacter shibae* DFL12 (40).

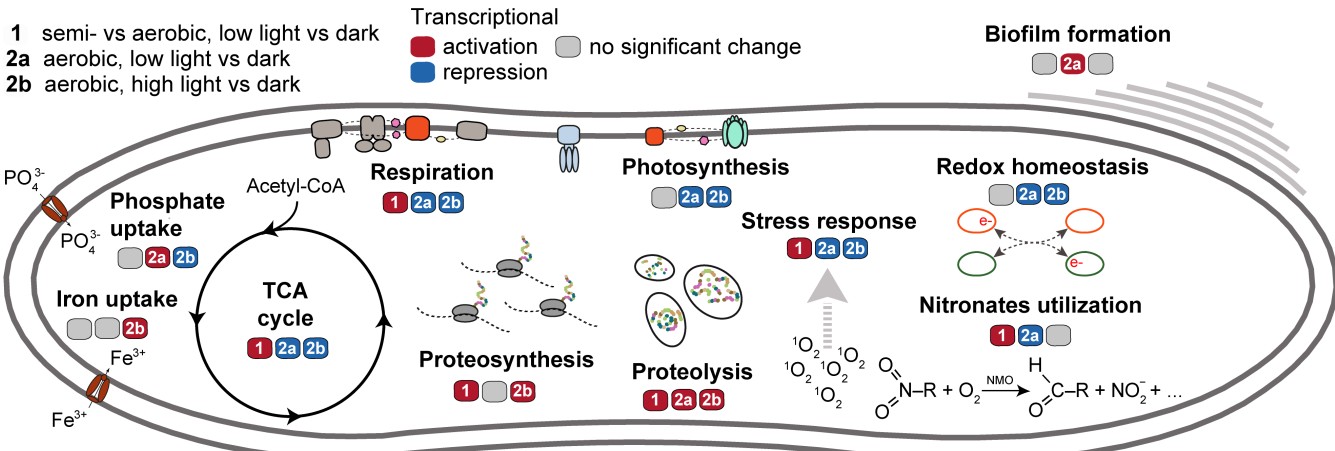

**FIG 5** Response of selected metabolic processes after transition to light. The model of metabolic processes summarizes their temporal expression changes into trends listed as either transcriptional repression (blue), activation (red), or no significant change (gray) in transcription. For experimental conditions, see the legend in the left upper corner.

Erlenmeyer flasks on orbital shakers. Semiaerobic cultures were grown in closed glass flasks bubbled with a mixture of nitrogen and air containing 10% $O_2$ (as well as approx. 200 ppm $CO_2$). Heterotrophic growth in the dark was monitored for 9 days with sampling for pigment and protein assays every second day starting day 3. To check the oxygen preferences of the strain, cells were grown inside 1% agar in a glass tube. Illumination for the dark–light transition and light intensity experiments was provided by a bank of Dulux L 55W/865 luminescent tubes (Osram GmbH, Munich, Germany, spectral temperature of 6,500 K) and delivered either continuously or using a 12-hour light–12-hour dark regime. The light intensity gradient from 30 to 500 µmol photon $m^{-2}$ $s^{-1}$ was created by placing the cultures at various distances from the light source. The cultures were protected from the heat radiated by the lamp by a 30-mm-thick planar water filter. For each light intensity, samples for the pigment and protein assays were taken on days 5, 7, 9, and 10. Sampling for the dark-to-light switch was done 2 and 8 hours or 2, 4, and 8 hours after the shift to light. The control set was kept in the dark.

## Pigment and protein assays

The cells were collected by centrifugation (10,000 $\times$ $g$ for 5 min); the pellet was gently resuspended in 40-µL water and then extracted with 1-mL acetone:methanol (7:2 vol:vol). The pigment extracts were analyzed using a high-performance liquid chromatography system Nexera LC-40 HPLC system (Shimadzu Inc., Tokyo, Japan) equipped with the heated (40°C) Phenomenex Luna 3µ C8(2) 100 Å and a diode-array UV–VIS detector as described earlier (41). Protein content was determined spectrophotometrically using a Lowry assay kit (Merck, Germany).

## RNA sequencing and transcriptome analysis

Biological triplicates were sampled. Cells were harvested by centrifugation. Pellets were resuspended in a 1-mL PGTX extraction solution (46) and immediately frozen in liquid nitrogen. RNA was extracted and processed as described earlier (41). Briefly, samples were incubated at 95°C for 5 min and immediately placed on ice for 10 min. After the addition of 800 mL chloroform, the extraction mix was centrifuged to promote phase separation. The aqueous phase was then retrieved and mixed with an equal volume of chloroform, centrifuged, and retrieved again. RNA was precipitated with isopropanol overnight at −20°C, recovered by centrifugation, washed with 70% ethanol, air-dried, and finally dissolved in an appropriate volume of sterile nuclease-free water. The RNeasy kit (Qiagen, the Netherlands) was used for purification according to the manufacturer's manual. The first digestion of genomic DNA was performed on the column, using DNase I (Qiagen, the Netherlands) according to the manufacturer's protocol. Total RNA was eluted in 88-mL RNase-free $H_2O$, and the second DNase I digestion was made in solution, followed by a second RNeasy purification step, which included an additional washing step with 80% ethanol done before elution with 30-mL RNase-free water. Samples were tested for genomic DNA contamination by using RNA directly as a template for PCR. Possible contaminating DNA was removed using the TURBO DNA-free kit (Ambion) according to the manufacturer's protocol. Libraries were generated according to Shishkin et al. (47) including rRNA removal with the RiboZero Kit (Illumina Inc., San Diego, CA, US). The library was sequenced on a NovaSeq 6000 (Illumina Inc., San Diego, CA, US) in the paired-end mode with 100 cycles in total using the FASTQ-mcf suite (https://github.com/ExpressionAnalysis/ea-utils, December 2021). The image analysis and base calling were performed using the Illumina pipeline v 1.8 (Illumina, San Diego, CA, USA). Raw reads were processed, and differential gene expression was assessed as described before (41). Low-quality bases (Phred score <30) and Illumina adapters were clipped. Briefly, quality-filtered reads were mapped to the AP64$^T$ genome (NCBI GenBank accession GCF_000695095.2) using bowtie2 (48). FeatureCounts was used to assess the number of reads per gene (49). Low normalization and identification of significantly differentially regulated genes (false discovery rate <0.01 and absolute $\log_2$ FC >1) were performed with edgeR (50). The heatmap was generated with the package pheatmap. Hierarchical

clustering based on the Euclidian distance of $\log_2$ FC data was used to cluster genes. Binding sites for PpsR were obtained from the Prodoric database (51).

## BChl fluorescence

The sampled cell suspension was diluted into 4 mL of fresh medium to 100-nM BChl *a* concentration. Following dark adaptation for 5 min, fluorescence induction was elicited by a single-turnover saturating flash length 50 µs, 0.33 mol photon $m^{-2}$ $s^{-1}$ delivered by an array of cyan 505 nm Luxeon Rebel diodes, and the kinetics were recorded using the FL-3000 fluorometer (Photon Systems Instruments Ltd., Czech Republic) at λ > 850 nm with a 100-ns resolution [for details, see Kaftan et al. (52)]. BChl *a* fluorescence decay following the single-turnover saturating flash was monitored by logarithmically spaced 1-µs flashlets.

## ACKNOWLEDGMENTS

The authors thank Astrid Dröge (Helmholtz Centre for Infection Research, Braunschweig, Germany) for her help with RNA sequencing library preparation and Alastair T. Gardiner for language correction.

This research was supported by the Czech Science Foundation project PhotoGemm+ GX19-28778X. M.K.S. was supported by the MSCA Fellowships CZ program (OP JAK) financed by the Czech Ministry of Education. Finalization of this study was supported by the OP JAK project Photomachines reg. no. CZ.02.01.01/00/22_008/0004624.

## AUTHOR AFFILIATION

[1]Laboratory of Anoxygenic Phototrophs, Institute of Microbiology of the Czech Acad Sci, Třeboň, Czechia

## AUTHOR ORCIDs

Karel Kopejtka http://orcid.org/0000-0001-5412-4541
Jürgen Tomasch http://orcid.org/0000-0002-3914-2781
Sahana Shivaramu http://orcid.org/0000-0003-4037-8051
Mohit Kumar Saini http://orcid.org/0000-0002-2409-0923
David Kaftan https://orcid.org/0000-0003-0932-0986
Michal Koblížek http://orcid.org/0000-0001-6938-2340

## FUNDING

| Funder | Grant(s) | Author(s) |
| --- | --- | --- |
| Grantová Agentura České Republiky (GAČR) | GX19-28778X | Karel Kopejtka |
| | | Jürgen Tomasch |
| | | Sahana Shivaramu |
| | | Mohit Kumar Saini |
| | | David Kaftan |
| | | Michal Koblížek |
| Ministerstvo Školství, Mládeže a Tělovýchovy | CZ.02.01.01/00/22_008/0004624 | Karel Kopejtka |
| | | Jürgen Tomasch |
| | | Sahana Shivaramu |
| | | Mohit Kumar Saini |
| | | David Kaftan |
| | | Michal Koblížek |

## AUTHOR CONTRIBUTIONS

Karel Kopejtka, Data curation, Investigation, Visualization, Writing – original draft, Writing – review and editing | Jürgen Tomasch, Data curation, Investigation, Writing – original draft, Writing – review and editing | Sahana Shivaramu, Investigation | Mohit Kumar Saini, Investigation | David Kaftan, Investigation | Michal Koblížek, Conceptualization, Funding acquisition, Investigation, Writing – original draft, Writing – review and editing

## DATA AVAILABILITY

RNA sequencing data are publicly available at the NCBI gene expression omnibus database under accession number GSE253349-GSE253351.

## ADDITIONAL FILES

The following material is available online.

### Supplemental Material

**Figure S1 (mSystems00706-24-s0001.tif).** Transcriptional response to light under different $O_2$ concentrations.
**Figure S2 (mSystems00706-24-s0002.tif).** The effect of light intensity on the activity of reaction center with two rings of light-harvesting antennae.
**Tables S1 to S3 (mSystems00706-24-s0003.xlsx).** Transcriptomes.
**Legends (mSystems00706-24-s0004.docx).** Legends for Figures S1 and S2.

### Open Peer Review

**PEER REVIEW HISTORY (review-history.pdf).** An accounting of the reviewer comments and feedback.

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
