## [Reviewer comments · mSystems]

Minimal transcriptional regulation of horizontally transferred photosynthesis genes in phototrophic bacterium *Gemmatimonas phototrophica*

Karel Kopejtka, Jürgen Tomasch, Sahana Shivaramu, Mohit Saini, David Kaftan, and Michal Koblizek

Corresponding Author(s): Michal Koblizek, Mikrobiologicky ustav Akademie ved Ceske republiky

Review Timeline:

Submission Date:	May 23, 2024
Editorial Decision:	June 24, 2024
Revision Received:	July 26, 2024
Accepted:	August 1, 2024

Editor: Chris Greening

Reviewer(s): Disclosure of reviewer identity is with reference to reviewer comments included in decision letter(s). The following individuals involved in review of your submission have agreed to reveal their identity: Shuhei Yabe (Reviewer #1); Paulina Prondzinsky (Reviewer #2)

Transaction Report:

DOI: <https://doi.org/10.1128/msystems.00706-24>

Re: mSystems00706-24 (Minimal transcriptional regulation of horizontally transferred photosynthesis genes in phototrophic bacterium *Gemmatimonas phototrophica*)

Dear Dr. Michal Koblížek:

Revision Guidelines

Sincerely,
Chris Greening
Editor
mSystems

Reviewer #1 (Comments for the Author):

Reviewer Report for Manuscript mSystems00706-24

This manuscript reveals that *Gemmatimonas phototrophica*, known for acquiring its photosynthesis gene cluster (PGC) through horizontal gene transfer from purple bacteria, does not exhibit the typical oxygen and light regulation systems common to purple bacteria. This is a significant finding from the perspective of microbial evolution. However, several improvements and corrections

are necessary to enhance the clarity and impact of the manuscript.

Introduction

The introduction provides a comprehensive background on photosynthesis in bacteria and horizontal gene transfer. However, the flow of the Introduction section is not smooth. In the current manuscript, general information about *G. phototrophica* starts from line 76, followed by previous research on BChl production in response to oxygen and light, and then the topic of horizontal gene transfer and the hypothesis.

To improve the flow, it would be better to detail the horizontal gene transfer of the entire PGC from purple bacteria in the general information section about *G. phototrophica*. Following this, introduce previous research on the PGC and BChl production in *G. phototrophica*, and then explain the rationale for your hypothesis and state your objectives. This structure would make the importance of your research clearer.

- Line 53: *Vulcanimicrobiota* should be written as *Vulcanimicrobiota* (formerly known as *Ca. Eremiobacteriota* or WPS-2).

Materials and Methods

Throughout the Materials and Methods section, the sampling timings for the analyses are mentioned in the Results section. However, these details should be described more specifically in the Methods section (e.g., Lines 115-116, 120-122, 139-143). Additionally, the light/dark regime should be included in the Methods section.

- Line 104-105: Please provide a brief description of the optimized medium used.
- Line 109-110: According to Shivaramn et al., 2023, the adjustment to 10% O₂ was achieved using a mixture of 90% N₂ and 10% O₂, not Air. Could the use of Air in your experiment introduce CO₂ and other trace gases that might affect the results?
- Line 118-119: Include the name of the column used for HPLC.
- Line 122: Briefly describe the RNA extraction method, including at least the name of the kit used.
- Line 120-122: The number of biological replicates for the RNA-seq experiment is not explicitly mentioned. Please specify the number of replicates (e.g., n=3) in the Methods section.
- Line 139-143: Please provide a brief description of the sample preparation method.

Results and Discussion

The manuscript mentions that oxygen concentrations were adjusted using a mix of air and N₂. However, using air could introduce CO₂ and other trace gases, which might affect the expression of photosynthesis-related genes in *G. phototrophica*. It would be beneficial to address whether these factors were considered and discuss their potential impact on the experimental results.

The study uses a 10% oxygen concentration as the semiaerobic condition for this strain. However, if *G. phototrophica* can grow at lower oxygen concentrations, it would be intriguing to explore the impact of such low oxygen levels on the expression profiles of photosynthesis-related genes. Investigating gene expression at oxygen concentrations near the lower limit of its growth range could provide deeper insights. If data on this are available, including and discussing it would make the study more comprehensive and meaningful.

- Lines 342-344: The manuscript suggests that *G. phototrophica* may thrive in environments characterized by reduced oxygen and lower light, such as freshwater lake metalimnia. This is particularly interesting, as the Introduction mentions its distribution in lakes. If there are previous studies showing its predominance in metalimnia, it would be beneficial to reference those and discuss the ecological aspects in more detail. This additional context would enhance the ecological and evolutionary discussion in the manuscript.

Reviewer #2 (Comments for the Author):

Summary

The authors present growth and transcriptomic responses of *Gemmatimonas phototrophica*, an aerobic anoxygenic phototroph, to different light and oxygen conditions and combinations of those. The organism is of special interest as it is the only known phototroph within the Gemmatimonadota and acquired all of its photosynthetic machinery via HGT.

The study sheds light on the regulatory patterns of the photosynthetic gene cluster in a variety of different growth conditions, including two different oxygen concentrations and 5 different light intensities with two different light cycles.

This is an important study on the regulatory mechanisms of genes acquired via HGT and the transcriptomic analyses of responses to abiotic factors. However, due to the analysis of both oxygen and light effects as well as temporal shifts, the distinction between growth conditions needs to be clarified in parts of the manuscript.

General Comments

1. Many different oxygen and light conditions were tested (two oxygen concentrations in two different light conditions (?) and five different light conditions each in diurnal cycles and under continuous light) and in some cases it is unclear in the figures and/or the text which conditions are being compared. For example, it is somewhat unclear what shift exactly is being looked at in Figure 1B (line 178): is it the difference between semiaerobic and aerobic (seems so from the text) or is it a shift from dark to light (seems so from the figure legend)? fold change positive means upregulation in semiaerobic? It took me some time to connect the text to the Figure and it is a lot clearer when looking at Supplementary Figure S1. You might want to consider moving panel B in addition to Supplementary Figure S1 to a separate figure in the text for clarity.

2. In some sections the language needs some revision for clarity (minor comments below).

3. As the study is based on those genes acquired through HGT, could you mark them as such in Figures 2 and 3 just like in Supplementary Figure S1?

4. A lot of the comparison relies on BChl a:protein ratios. While there is a detailed description for pigment extraction and analysis, it is unclear from the methodology section how protein content was quantified. How exactly were these ratios calculated? Did you measure the cellular protein content for cells in each growth condition? Biomass composition is known to vary (<https://doi.org/10.3390/molecules25122834>) and should be analysed in order to use the BChl a:protein ratios for comparison.

5. Make a clearer distinction between previous work (phylogeny) and presented work (expression) in your final paragraph (lines 334) and expand on contextualization of observed regulation patterns. "The gene regulation mechanisms of the original host were probably not transferred [...]" (line 337) is a very vague statement for the final paragraph of the conclusions.

6. Supplementary Figure S2 is a great visual summary of the findings of this study and should be included in the main text.

Minor in-line comments:

line 39: missing word "lacks [...] which regulate [the] synthesis"

line 62 - 70: "These photoheterotrophs fully adapted to oxic conditions and harvest light energy and grow in the presence of oxygen." and "Therefore, AAP species rapidly downregulate expression of photosynthesis genes when illuminated" are contradictory. It is clear that BChl synthesis can be shifted to the night from the following sentence, but please expand on how photosynthesis can be carried out when those genes are downregulated in light? Or do you mean that PS is downregulated in the presence of oxygen?

line 101: Add how this main objective will be achieved (main analytical methodologies used).

line 150: "gene" → "genes"

line 152: Add the Agar cultivation part to the methodology section, I thought all of the cultures were grown in liquid medium?

line 157: The ratio of BChl a per protein is the same, but does the overall protein content (biomass composition) change between the two oxygen conditions? Did you look at whether there is a shift in the composition to support this statement?

line 185: Membrane transport appears as up and down regulated genes. The details of DEGs can be found in the Supplementary Tables, however it might be worth it to include details for those found in both directions in the text.

line 216: This is the first time the reader is introduced to "RC-dLH". Add a minor description such as "antenna complex".

line 215-220: Is this data shown somewhere? If not, please provide it in the supplementary materials.

line 226: remove "them"

line 264: It is an interesting observation that DEGs that are upregulated follow two distinct temporal patterns, while those being downregulated seem to follow only one. Can you expand on this a bit or refer to it in the discussion section?

line 270: "and all"?

line 283: remove "In"

line 285: add word "[they] produce pigment under lower..."

line 332: Can you give an explanation as to why none of the ROS defense genes were differentially expressed, especially

comparing low O2 vs higher O2 growth?

Figure comments:

Figure 1:

Add a scale bar in the inset of Figure 1A.

For Cluster 2 the DEGs add up to 11, but 13 are indicated in parentheses.

For Cluster 4 the DEGs add up to 15, but 16 are indicated in parentheses.

line 496: Rephrase "For normalization of these samples 496 were used samples kept in dark." → "For normalization samples kept in the dark were used." (?)

Figure 2:

Add "diurnal" and "continuous" within the respective panel of Figure 2 for easier comprehension?

Figure 3:

Panel A: Numbers of DEGs in clusters do not add up. I understand that for bigger clusters it might take up too much space to include all of them, however for smaller clusters you could consider adding the missing ones.

Panel B: Can you match the width of the heatmap cells to the width on the operon?

Reviewer Report for Manuscript mSystems00706-24

This manuscript reveals that *Gemmatimonas phototrophica*, known for acquiring its photosynthesis gene cluster (PGC) through horizontal gene transfer from purple bacteria, does not exhibit the typical oxygen and light regulation systems common to purple bacteria. This is a significant finding from the perspective of microbial evolution. However, several improvements and corrections are necessary to enhance the clarity and impact of the manuscript.

Introduction

The introduction provides a comprehensive background on photosynthesis in bacteria and horizontal gene transfer. However, the flow of the Introduction section is not smooth. In the current manuscript, general information about *G. phototrophica* starts from line 76, followed by previous research on BChl production in response to oxygen and light, and then the topic of horizontal gene transfer and the hypothesis.

To improve the flow, it would be better to detail the horizontal gene transfer of the entire PGC from purple bacteria in the general information section about *G. phototrophica*. Following this, introduce previous research on the PGC and BChl production in *G. phototrophica*, and then explain the rationale for your hypothesis and state your objectives. This structure would make the importance of your research clearer.

- **Line 53:** Vulcanimicrobiota should be written as *Vulcanimicrobiota* (formerly known as *Ca. Eremiobacteriota* or WPS-2).

Materials and Methods

Throughout the Materials and Methods section, the sampling timings for the analyses are mentioned in the Results section. However, these details should be described more specifically in the Methods section (e.g., Lines 115-116, 120-122, 139-143). Additionally, the light/dark regime should be included in the Methods section.

- **Line 104-105:** Please provide a brief description of the optimized medium used.
- **Line 109-110:** According to Shivaramn et al., 2023, the adjustment to 10% O₂ was achieved using a mixture of 90% N₂ and 10% O₂, not Air. Could the use of Air in your experiment introduce CO₂ and other trace gases that might affect the results?
- **Line 118-119:** Include the name of the column used for HPLC.
- **Line 122:** Briefly describe the RNA extraction method, including at least the name of the kit used.

- **Line 120-122:** The number of biological replicates for the RNA-seq experiment is not explicitly mentioned. Please specify the number of replicates (e.g., n=3) in the Methods section.
- **Line 139-143:** Please provide a brief description of the sample preparation method.

Results and Discussion

The manuscript mentions that oxygen concentrations were adjusted using a mix of air and N₂. However, using air could introduce CO₂ and other trace gases, which might affect the expression of photosynthesis-related genes in *G. phototrophica*. It would be beneficial to address whether these factors were considered and discuss their potential impact on the experimental results.

The study uses a 10% oxygen concentration as the semiaerobic condition for this strain. However, if *G. phototrophica* can grow at lower oxygen concentrations, it would be intriguing to explore the impact of such low oxygen levels on the expression profiles of photosynthesis-related genes. Investigating gene expression at oxygen concentrations near the lower limit of its growth range could provide deeper insights. If data on this are available, including and discussing it would make the study more comprehensive and meaningful.

- **Lines 342-344:** The manuscript suggests that *G. phototrophica* may thrive in environments characterized by reduced oxygen and lower light, such as freshwater lake metalimnia. This is particularly interesting, as the Introduction mentions its distribution in lakes. If there are previous studies showing its predominance in metalimnia, it would be beneficial to reference those and discuss the ecological aspects in more detail. This additional context would enhance the ecological and evolutionary discussion in the manuscript.

Re: mSystems00706-24R1 (Minimal transcriptional regulation of horizontally transferred photosynthesis genes in phototrophic bacterium *Gemmatimonas phototrophica*)

Dear Dr. Michal Koblizek:

Your manuscript has been accepted, and I am forwarding it to the ASM production staff for publication. Your paper will first be checked to make sure all elements meet the technical requirements. ASM staff will contact you if anything needs to be revised before copyediting and production can begin. Otherwise, you will be notified when your proofs are ready to be viewed.

Sincerely,
Chris Greening
Editor
mSystems